# Preparation of Chemically Modified Lignin-Reinforced PLA Biocomposites and Their 3D Printing Performance

**DOI:** 10.3390/polym13040667

**Published:** 2021-02-23

**Authors:** Seo-Hwa Hong, Jin Hwan Park, Oh Young Kim, Seok-Ho Hwang

**Affiliations:** Materials Chemistry & Engineering Laboratory, School of Polymer System Engineering, Dankook University, Yongin, Gyeonggi-do 16890, Korea; seohwa8766@naver.com (S.-H.H.); shssh123@gmail.com (J.H.P.); kyobon@dankook.ac.kr (O.Y.K.)

**Keywords:** poly(lactic acid), lignin, maleic anhydride, chemical modification, 3D printing filament

## Abstract

Using a simple esterification reaction of a hydroxyl group with an anhydride group, pristine lignin was successfully converted to a new lignin (COOH-lignin) modified with a terminal carboxyl group. This chemical modification of pristine lignin was confirmed by the appearance of new absorption bands in the FT-IR spectrum. Then, the pristine lignin and COOH-lignin were successfully incorporated into a poly(lactic acid) (PLA) matrix by a typical melt-mixing process. When applied to the COOH-lignin, interfacial adhesion performance between the lignin filler and PLA matrix was better and stronger than pristine lignin. Based on these results for the COOH-lignin/PLA biocomposites, the cost of printing PLA 3D filaments can be reduced without changing their thermal and mechanical properties. Furthermore, the potential of lignin as a component in PLA biocomposites adequate for 3D printing was demonstrated.

## 1. Introduction

Additive manufacturing is redefining manufacturing paradigms and has evolved rapidly in recent years. This technology allows for the production of customized parts, prototyping, and material design [1,2]. Although there are several technologies, fused deposition modeling (FDM), one of the most used 3D printing technologies, has attracted significant interest because of its low cost, ease of use, and large availability [3]. The most frequently used commercial materials for 3D printing by FDM are acrylonitrile-butadiene-styrene terpolymer (ABS) and poly(lactic acid) (PLA). Nevertheless, PLA is more widely used, due to issues concerning environmental pollution [4].

PLA is a versatile biopolymer polymerized from a renewable agricultural-based monomer, 2-hydroxy propionic acid (lactic acid), which is synthesized by fermentation of starch-rich materials like sugar beets, sugarcanes, and corn [5,6]. Therefore, this material is a promising and sustainable alternative to petroleum-based synthetic polymers used for a wide range of commodity applications since it can be processed in a similar manner to polyolefins, using the same processing machinery. Nonetheless, the inherent drawbacks of PLA, such brittleness, low heat resistance, high-cost, and slow crystallization, have impeded its widespread use in commercial applications.

To provide a solution to the aforementioned drawbacks, many interesting strategies, such as plasticization [7], block copolymerization [8,9], and compounding with other tougher polymers or elastomers have been utilized [10,11,12,13,14,15,16,17,18]. Another cost-effective approach to improve the mechanical resistance of PLA at high temperatures is the addition of filler materials to form biocomposites or nanocomposites. Recently, several types of lignin have been incorporated into PLA matrixes by melt-mixing techniques for the development of sustainable biocomposites [19,20,21,22]. Evenly applied in the chemically modified lignin, PLA-based biocomposites often show lower mechanical properties, like tensile strength and elongation at break, compared to neat PLA, due to their poor dispersion and adhesion. Several studies were carried out regarding incorporating lignin in the PLA matrix to form composite filaments for 3D printing, making it evident that lignin’s addition often leads to a decrease in tensile strength and elongation [23,24,25,26,27,28,29,30]. Filaments with poor properties will be the death of successful commercialization efforts. Therefore, the development of new lignin-based biocomposite materials for the production of polymeric filaments for FDM is still a challenge.

In this study, we focus on lignin introduced into FDM printing filaments as filler to reduce PLA filament cost. Since, to the best of our knowledge, pristine lignin without any chemical modification cannot aid in improving the physical properties, we selected a simple esterification reaction with maleic anhydride to prepare chemically modified lignin and improve its surface polarity. The carboxy group on the lignin surface is expected to exhibit better adhesion to the PLA matrix, due to its strong hydrogen bonding ability and PLA chain grafting covalently on the lignin surface during processing under high temperatures. Melt blending of PLA with a chemically modified lignin should enable 3D printed filament production. Therefore, the objective of this study is to characterize the thermal, mechanical, and morphological properties of the resulting lignin-based PLA biocomposites, demonstrating their suitability as filament materials for FDM 3D printing.

## 2. Materials and Methods

### 2.1. Materials

PLA pellets (trade name: Ingeo 2003D; *M*w (g/mol) = 181,744 [31]; melt flow index (MFI): 6.0 g/10 min; specific gravity: 1.24) was obtained commercially from NatureWorks LLC (Minnetonka, NM, USA). Organosolv lignin (pH = 6.9–7.1; ash <16%) was obtained from BOC Sciences (Shirley, NY, USA). Maleic anhydride and other solvents used in this work were purchased from SAMCHUN Chemicals (Seoul, Korea). 

### 2.2. Preparation of Carboxylated Lignin (COOH-Lignin)

First, lignin (10 g) was dispersed in dimethylforamide (DMF, 150 mL) using a homogenizer (T10 basic ULTRA-TURRAX; IKA-Werke GmbH &Co. KG, Staufen, Germany) at 25 °C. Then, maleic anhydride (70 g, 0.71 mol) was added to the solution and stirred vigorously for 6 h at 120 °C. After the crude mixture was cooled to 25 °C, the modified lignin was obtained by centrifugation and washed thoroughly with ethanol and deionized in water several times. Then, the sample was kept in vacuo at 50 °C for a day to dry completely.

### 2.3. 3D Printing

The monofilaments with diameters of about 1.75 mm were extruded using the Wellzoom 3D desktop filament extruder (Shenzhen Mistar Technology Co., Ltd., Shenzhen, China) at 210 °C. A FDM-type 3D printer (model: Ender 3, Shenzhen Creality 3D Technology Co., Ltd., Shenzhen, China) was used to print the prepared filaments into the sample at a printing temperature of 220 °C.

### 2.4. Equipment and Experiments 

FT-IR spectra were recorded on a Nicolet iS10 spectrophotometer (Thermo Scientific Co., Ltd., Waltham, MA, USA) over a range of 4000 to 600 cm^−1^. To measure the thermal behavior of samples, differential scanning calorimetry (DSC) analysis was carried out using the DSC Q2000 apparatus (TA Instruments, New Castle, DE, USA). The scan was performed at the heating rate of 20 °C/min under N_2_ gas atmosphere. Tensile properties were measured according to ASTM D638-10 at 25 °C using an Instron universal testing machine (LSK 100K; Lloyd Instruments Ltd., West Sussex, UK), equipped with a 10 kN load cell at a constant crosshead speed of 5 mm/min. The tensile properties were evaluated from averages of at least five parallel tests. After preparing the specimen sheet by the typical compression molding process, the tensile test specimen was cut by specimen cutting die into a dumbbell shape (ASTM D638, Type V). The fractured surface and profile morphology of the composites were imaged and analyzed using an EM-30 scanning electron microscope (SEM; Coxem, Deajeon, Korea) at 10 kV.

## 3. Results and Discussion

The compatibility of lignin with commercial polymers is generally limited because it has a complicated steric hinderance induced from the crosslinking and disorder of the molecular structure. Thus, chemical modification of lignin can provide an important key strategy to further expand lignin applications [32]. Among the many important functional groups (phenolic hydroxyl, aliphatic hydroxyl, carbonyl, alkyl aryl ether, biphenyl, diaryl ether, phenylpropane, guaiacyl, syringyl, etc.) [33,34,35,36] in lignin, the hydroxy group can react with the anhydride group through a well-known esterification reaction (Scheme 1) because the carbon in the anhydride moiety is a strong electrophile, largely due to the fact that the substantial ring strain is relieved when the ring opens upon nucleophilic attack. Under this organic chemistry, maleic anhydride was chosen as the lignin modifier, and we assumed that the carboxy group covalently linked to the lignin can serve as a hydrogen bonding donor and acceptor, as well as a reaction site capable of transesterification with PLA.

To obtain information on chemical changes resulting from the chemical modification of lignin, we used FT-IR spectroscopy. Figure 1 shows the FT-IR absorption spectra based on the molecular vibrations for COOH-lignin and pristine lignin. In the case of pristine lignin, a broad band attributed to aromatic and aliphatic hydroxyl groups in the lignin molecules was detected around 3400–3500 cm^−1^. In addition, the appearance of two bands centered around 2929 and 2848 cm^−1^ was clearly seen, arising from CH_2_ stretching modes in aromatic methoxy groups and in methyl and methylene groups of side chains. We also found three distinct absorption bands originating from the aromatic skeleton vibration in the lignin molecules around 1586, 1496, and 1451 cm^−1^ [37]. After chemical modification of pristine lignin, the appearance of two new bands around 1722 and 1127 cm^−1^ in the spectrum evidences the formation of ester groups containing one C=O bond and two C–O bonds on the lignin surface. Finally, the vinyl group-related absorption band-like shoulder around 1617 cm^−1^ indicated that maleic anhydride successfully reacted on the surface of the pristine lignin. The FT-IR results indicated that the simple esterification reaction of maleic anhydride on the surface of the lignin particles was successful. 

The effect of the presence of pristine lignin and COOH-lignin on the thermal behavior of the PLA matrix was studied using DSC. Table 1 presents thermal characteristics of the melting temperature (*T*_m_) observed from the second heating cycle and glass transition temperature (*T*_g_) for pristine lignin- or COOH-lignin-reinforced PLA biocomposites. On the basis of Table 1, compatibility between the PLA matrix and lignin domain in the biocomposites should be inferred. The presence of lignin in the PLA matrix did not significantly affect the melting-point depression phenomenon, which decreased from 152.2 to 149.2 °C with increasing content of pristine lignin, consistent with the behavior of lignin as a filler and not a nucleating agent for all the studied PLA-based biocomposites. On the DSC thermograms, the glass transition temperatures of COOH-lignin-reinforced PLA biocomposites showed similar trends to that of pristine lignin-reinforced PLA biocomposites, ranging from 60.3 to 63.4 °C. Further, the sharp *T*_g_ transition occurred in all biocomposites. Sometimes, a broad *T*_g_ transition in a polymer composite system is caused by the complex molecular structure that leads to multiple relaxation phases within the same transition temperature range [38]. Therefore, even the lignin was chemically modified by maleic anhydride on its surface, though it did not significantly affect the thermal characteristics of the PLA matrix in the biocomposite.

Dispersibility and interfacial adhesion of filler to the polymer matrix are key factors in determining the final physical properties of a filler-reinforced composite. To evaluate the dispersibility of lignin in the PLA matrix and interfacial adhesion, the morphological characteristics of cryo-fractured pristine lignin- or COOH-lignin-reinforced PLA biocomposites were analyzed using the SEM technique. Figure 2 and Figure 3 show typical SEM micrographs at the fracture surface of the pristine lignin- and COOH-lignin-reinforced PLA biocomposites containing 5, 10, 15, and 20 wt.% of each lignin, respectively. As shown in Figure 2, the fracture surfaces of the pristine lignin/PLA biocomposites were rougher than the COOH-lignin/PLA biocomposites in all ranges of lignin content. In addition, many voids and a wide gap morphology between the lignin domain and the PLA matrix can be easily seen, as evidenced by the observation of debonded areas at the fracture surface, revealing that interfacial adhesion at the interphase does not appear very intense. Somewhat different from the pristine lignin/PLA biocomposites, the COOH-lignin-reinforced PLA biocomposites were smooth and featureless and seemed to show stronger interfacial adhesion with the matrix through the presence of tight adhesive areas in the intermaterial without any room (Figure 3).

To obtain the maximum effect in the filler reinforced composite, the filler should have good dispersibility and interfacial adhesion with the polymer matrix. Here, we investigated the mechanical properties of lignin-reinforced PLA biocomposites following a chemical surface modification of the lignin. Figure 4 presents the dependence of tensile strength and modulus with lignin content for the PLA biocomposites with pristine lignin and COOH-lignin. As shown in Figure 4A, the tensile strength of the pristine lignin/PLA biocomposites decreased linearly with increasing pristine lignin content up to 20 wt.%. This declining trend in tensile strength of the PLA-based biocomposites likely results from poor dispersibility and interfacial adhesion between the pristine lignin domains and PLA matrix [30]. Meanwhile, with COOH-lignin in the PLA matrix, the tensile strength dropped slightly to 50 MPa up to 5 wt.% lignin and then stagnated with further addition of COOH-lignin, manifesting a better COOH-lignin wetting effect with PLA resin. The tensile modulus of the pristine lignin- or COOH-lignin-reinforced PLA biocomposites exhibited similar decreases until 15 wt.% lignin content [Figure 4B]. However, with a 20 wt.% applied to lignin content in the PLA matrix, the tensile modulus showed better values (about 5.0 GPa) compared to both the pristine lignin/PLA biocomposite and pure PLA resin. The above results indicate that COOH-lignin exhibits better dispersibility in the PLA matrix than pristine lignin.

3D printed objects fabricated using COOH-lignin-reinforced PLA biocompoite filaments through a single-screw filament extruder were used to demonstrate the 3D printing behavior and print quality. It was difficult to make the pristine lignin-reinforced PLA biocomposite filaments having a uniform diameter but, all COOH-lignin-reinforced PLA biocompoite filaments exhibited acceptable diameter tolerances (1.41 ± 0.072 mm) for the 3D printer used in this study (Figure 5). The filaments containing COOH-lignin were successfully used to prepare the objects, as illustrated in Figure 6A–E. A closer examination of the surface roughness at smaller scales were obtained from SEM images of the object surfaces [Figure 6a–e]. The surface of the filaments on the 3D printed objects became discernibly rougher and darker in color with increasing COOH-lignin content. This is due to the relatively lower melt strength of COOH-lignin-reinforced PLA biocomposites and the dark brown color of the lignin. However, the surface roughness of each line on the side of the 3D printed object fabricated from PLA biocomposite filament containing more than 15 wt.% COOH-lignin was too rough to bind layer-by-layer, showing a wide gap. This behavior is due to the decreasing melt flow from the nozzle and resultant inadequate adhesion between layers.

## 4. Conclusions

By utilizing the reaction of the free hydroxyl group in lignin with the anhydride group, we demonstrated the possibility of realizing surface-modified lignin decorated by carboxyl functional groups. Two different PLA-based biocomposite series with pristine lignin and COOH-lignin were prepared under melt-mixing conditions. The tensile strength of the PLA-based biocomposites indicated better performance of the COOH-lignin/PLA biocomposites than of the pristine lignin/PLA biocomposites, due to improved interfacial adhesion between the COOH-lignin surface and PLA matrix through hydrogen bonding. However, there were no significant differences in the tensile moduli for both PLA-based biocomposites. Interestingly, the tensile modulus of the PLA-based biocomposite containing 20 wt.% COOH-lignin increased suddenly. We also conclude that a COOH-lignin content of 10 wt.% is the most cost effective for 3D FDM filaments. Therefore, PLA-based biocomposites with COOH-lignin can be developed as a new category of polymer materials specially used for the FDM 3D printing method in industrial applications.

## Data Availability

The data presented in this study are available on request from the corresponding author.

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
