# Peer review of "Preparation of Chemically Modified Lignin-Reinforced PLA Biocomposites and Their 3D Printing Performance"

_polymers, 2021, doi:10.3390/polym13040667_

Round 1
Reviewer 1 Report
Dear Authors,
My comments are provided in the attached pdf file.

Author Response
<Reviewer 1>
The authors have converted pristine lignin to a new lignin called here (COOH-lignin), modified with a terminal carboxyl group using a simple esterification reaction of a hydroxyl group with an anhydride group. Then, the pristine lignin and COOH-lignin were successfully incorporated into a poly(lactic acid) (PLA) matrix by a typical melt-mixing process. The interfacial adhesion performance between the lignin filler and PLA matrix was better and stronger than pristine lignin. The authors have also concluded that the cost of printing PLA 3D filaments can be reduced without changing their thermal and mechanical properties.
Although it’s an interesting research, however, there are a couple of matters which need to be addressed before the manuscript can be accepted in “Polymers” and hence, I recommend minor revision with the following comments:
- Page 2, line 63-73: In the preparation method, both Mol and the equivalent weight (in gram) should be mentioned.
- The description of maleic anhydride was represented using by mole unit. But lignin was just represented the description as gram because it could not measure the molecular weight due to its poor solubility in GPC solvents.
- Page 4, line 150: “….biocomposites was analyzed using a SEM” should be: ….biocomposites was analyzed using the SEM technique.
- The term ‘was analyzed using a SEM’ was replaced by ‘was analyzed using the SEM technique’ in the main text.
- The authors have used the “melting technique” to make these 2 different bio-composites, I was wondering if these bio-composites could be produced via extrusion as well? Have the authors tried to test that?
- The biocomposite samples were prepared using by internal mixer (like as HAAKE internal mixer) in this study. We did not use the typical extruder such as single- or twin-screw extruder. However, we used the small desktop extruder (Wellzoom 3D desktop filament extruder) to prepare the extruded filament for applying the 3D FDM printing.
- Have the authors looked into the degradability of these bio-composites in lab-induced environments? And investigating how these new bio-composites affect this PLA characteristic?
- We did not carry out biodegradability test. In this study, we assumed that the prepared PLA-based biocomposite containing lignin should be biodegradable because the previous paper (Materials Science & Engineering B 251 (2019) 1144412) provided the biodegradability result for PLA/pristine lignin composites.
- One of the important factors for 3D printing is the flow conditions for this process, have the authors tried to calculate the flow rate and attribute it to these bio-composites and their characteristics? As this is quite of interest in the industry.
- Yes, we agree your comments for the flow condition in 3D printing process. In this article, we could not measure the flow rate because the 3D FDM printer used in this study has simple function to produce the object. We are coworking with other research group studying the transport phenomena of polymeric materials used in 3D printing and the detailed results for this concern will be submitted in a journal in near future.
- Would have been helpful to include a picture showing both the 3D printing materials, in this case, COOH-lignin-reinforced PLA bio-compoite filaments and the 3D printed material next to each other to give somesense the reader (refer to: Liu, L., Lin, M., Xu, Z., and Lin, M. (2019). "Polylactic acid-based wood-plastic3D printing composite and its properties," BioRes. 14(4), 8484-8498.)
- In this article, we fabricated simple shape objects using by simple functional 3D printer to test the possibility of application on FDM 3D printing process about lignin-reinforced PLA biocomposite. Thus, the picture of the 3D printed objects was represented in Figure 5(A-E) and the biocomposite filament samples were showed in Figure 4, newly.
Reviewer 2 Report
This manuscript presented a work about the modification and characterization of lignin for use as filler in PLA biocomposites applied to 3D print. The work has some potential. However several points listed below need to improved.
Introduction section: please add more studies related to PLA/lignin composites properties.
Section 2.1: what is the mean particle size of the organosolv lignin used. Also correct the typo, is organosolv instead of “organosolve”.
FTIR results: I can not indentify the bands described by the authors in the FTIR spectrum. Please indicate each band in Figure 1. In addition, I suggest improve the discussion in this section. There are others modification in the COOH-lignin spectrum that are not discussed by the authors.
A NMR characterization of lignin before and after modification would be very appreciated.
Table 1: the results presented in Table 1 seem to be the same for all biocomposites. If possible, add the standard deviation for all values.
Lines 157-160: the discussion in this part is a little confusing. Please improve the discussion in this part.
Figure 2: I suggest add micrographs that show the polymer/filler interface with higher magnification. It is very hard to see the interface between the to materials in Figure 2.
Figure 4: the authors must add the figures of PLA/pristine lignin composites the better comparison. In addition, the authors must comment the results obtained when PLA/pristine lignin composites were printed.
Author Response
<Reviewer 2>
This manuscript presented a work about the modification and characterization of lignin for use as filler in PLA biocomposites applied to 3D print. The work has some potential. However several points listed below need to improved.
- Introduction section: please add more studies related to PLA/lignin composites properties.
- Yes, we wrote new sentence relating to research results of PLA/lignin composite in the introduction section and several references were added in reference part.
- Section 2.1: what is the mean particle size of the organosolv lignin used. Also correct the typo, is organosolv instead of “organosolve”.
- We did not check the mean particle size of the lignin because there is no information of particle size in technical data sheet supplied from BOC science Ltd. After receiving of revision letter for this article, we just confirmed that the particle size as well as its shape of the organosolv was very diverse from 1 μm to 60 μm. Thus, we think that the particle size is not important in this study. And the term ‘organosolve’ was replaced by ‘organosolv’.
- FTIR results: I cannot identify the bands described by the authors in the FTIR spectrum. Please indicate each band in Figure 1. In addition, I suggest improve the discussion in this section. There are others modification in the COOH-lignin spectrum that are not discussed by the authors.
- We modified Figure 1 according to the reviewer’s comment. I agree with reviewer opinion for another absorption bands for COOH-lignin in FTIR spectrum. However, we only focused the where the chemical modification of pristine lignin was successfully done or not. Indeed, the other absorption bands for COOH-lignin in the spectrum could not figure out their identities because we could not find the information about those.
- A NMR characterization of lignin before and after modification would be very appreciated.
- Because the lignin is not soluble to typical NMR solvents, we cannot characterize the pristine lignin and COOH-lignin by NMR technique.
- Table 1: the results presented in Table 1 seem to be the same for all biocomposites. If possible, add the standard deviation for all values.
- We added the value of standard deviation for all values in Table 1.
- Lines 157-160: the discussion in this part is a little confusing. Please improve the discussion in this part.
- We modified the sentence pointed out by reviewer.
- Figure 2: I suggest adding micrographs that show the polymer/filler interface with higher magnification. It is very hard to see the interface between the to materials in Figure 2.
- Sorry, I don’t agree the reviewer’s comment. It is possible to recognize the difference of interface between lignin filler and matrix at the images in Figure 2. From the Figure 2, it can enough confirm that COOH-lignin was better than the pristine lignin in point of view the filler dispersitbility and adhesion performance.
- Figure 4: the authors must add the figures of PLA/pristine lignin composites the better comparison. In addition, the authors must comment the results obtained when PLA/pristine lignin composites were printed.
- We did not use the PLA biocomposites filament containing pristine lignin to fabricate 3D printed object because it was very difficult to make the filaments having uniform diameter. This explanation was mentioned in main text.
Reviewer 3 Report
Dear Authors,
in your interesting manuscript, the following points should be added/changed to further improve it:
- References: Is there really only one reference from 2020 and none from 2019? Please check again whether there may recent publications fitting to your study.
- End of page 1: "Even applied the chemically modified lignin" - which modification, the same as in your study?
- Generally, the introduction has in my opinion a small problem. You start with the aim to improve the mechanical and thermal properties of PLA ... and then offer a solution to reduce the material costs. Please try making clear that both points are interesting and that blending PLA with other materials should at least not decrease its mechanical and thermal properties. Besides, are the costs really the only argument to add lignin, or are there also ecological reasons?
- 2.2: Please define DMF. Is this procedure completely your own idea, or has it been reported before in the literature?
- 2.3: Please add the filament diameter, if possible with standard deviation. Was the pure PLA also extruded with the same instrument or bought? And please give the printing parameters which you used during preparation of the tensile test specimens.
- 2.4: Please mention the dimensions of the dumbbell shape samples for the tensile tests.
- Scheme 1: I would suggest writing "lignin" with small "l". And maybe it makes sense to use the "broad" form of figures here; with more space you could enlarge the lignin formula a little bit so that it is better visible.
- Table 1: Here you show values of up to 20 % lignin. What is missing before is an explanation how much lignin was maximally tested and to which amount PLA could be exchanged by lignin, without getting problems with the filament extruder.
- Below: What exactly is meant here with "biocomposites" - is this the blend in "any" shape, or the extruded filament, or the printed specimens?
- Fig. 2: Please think about using (a)-(h) for the sub-images. And please add the scale bars in a larger version so that they are well visible.
- Fig. 3b: The large value for 20 wt% is interesting - it's a pity that the difference is not significant. Maybe it is worth going on in this direction in the next study.
- Below Fig. 3, there is again the filament extruder mentioned, but the tensile strength was already measured using the 3D printed samples ... so this order doesn't seem to be ideal. Maybe Fig. 4 and the corresponding text should be shown before Fig. 3.
- Above Fig. 4 you mention "decreasing melt flow" for a larger amount of lignin - would it be supportive to increase the nozzle temperature?
Author Response
<Reviewer 3>
In your interesting manuscript, the following points should be added/changed to further improve it:
- References: Is there really only one reference from 2020 and none from 2019? Please check again whether there may recent publications fitting to your study.
- We attached new references published in recent from #23 to #30.
- End of page 1: "Even applied the chemically modified lignin" - which modification, the same as in your study?
- Especially, the sylilation on surface of lignin shows similar result.
- Generally, the introduction has in my opinion a small problem. You start with the aim to improve the mechanical and thermal properties of PLA ... and then offer a solution to reduce the material costs. Please try making clear that both points are interesting and that blending PLA with other materials should at least not decrease its mechanical and thermal properties. Besides, are the costs really the only argument to add lignin, or are there also ecological reasons?
- Because lignin is one of biodegradable biomass and has been used as industrial waste to generate heat, the high cost of PLA will be decreased by compounding with them. In this time, the characteristics of PLA should be maintained. Thus, the aim of this study is production of low-price PLA biocomposites having the enough properties for application of 3D FDM printing process.
- 2.2: Please define DMF. Is this procedure completely your own idea, or has it been reported before in the literature?
- We replaced the abbreviation (DMF) to full name as ‘dimethylforamide’. We think that this modification was done from my own concept.
- 2.3: Please add the filament diameter, if possible, with standard deviation. Was the pure PLA also extruded with the same instrument or bought? And please give the printing parameters which you used during preparation of the tensile test specimens.
- We added the diameter of filaments (1.41 ±072 mm) in main text. Yes, the pure PLA filament was extruded by the same instrument (Wellzoom 3D desktop filament extruder). The tensile test specimens were not made by 3D printing process. It was made using by specimen cutting die for a thin sheet fabricated by typical compression molding process.
- 2.4: Please mention the dimensions of the dumbbell shape samples for the tensile tests.
- The tensile test specimens were made by specimen cutting die according to ASTM D638 type V.
- Scheme 1: I would suggest writing "lignin" with small "l". And maybe it makes sense to use the "broad" form of figures here; with more space you could enlarge the lignin formula a little bit so that it is better visible.
- We modified the scheme 1 to replace from ‘L’ to ‘l’. We think that the size of scheme 1 is not small to understand the research concept.
- Table 1: Here you show values of up to 20 % lignin. What is missing before is an explanation how much lignin was maximally tested and to which amount PLA could be exchanged by lignin, without getting problems with the filament extruder.
- We will try how much lignin can be incorporated in PLA matrix and report those result in near future.
- Below: What exactly is meant here with "biocomposites" - is this the blend in "any" shape, or the extruded filament, or the printed specimens?
- Biocomposites are generally defined as eco-friendly composites. Biocomposite mentioned in this study consist of PLA as biodegradable polymer and lignin as biodegradable filler. Since the combined composite will be biodegradable, we used as 'biocomposite' term to the composites fabricated in this study.
- Fig. 2: Please think about using (a)-(h) for the sub-images. And please add the scale bars in a larger version so that they are well visible.
- We modified Figure 2 adding new scale bar.
- Fig. 3b: The large value for 20 wt% is interesting - it's a pity that the difference is not significant. Maybe it is worth going on in this direction in the next study.
- I agree with reviewer opinion.
- Below Fig. 3, there is again the filament extruder mentioned, but the tensile strength was already measured using the 3D printed samples ... so this order doesn't seem to be ideal. Maybe Fig. 4 and the corresponding text should be shown before Fig. 3.
- In this study, the samples for tensile test were prepared using by specimen cutting die for a thin sheet fabricated by typical compression molding process.
- Above Fig. 4 you mention "decreasing melt flow" for a larger amount of lignin - would it be supportive to increase the nozzle temperature?
- I am not sure. The increased nozzle temperature should be helpful to increase melt flow but, we concerned the extruded layer collapsing the shape due to low melt viscosity.
Round 2
Reviewer 2 Report
After corrections the manuscript reads well. I suggest publication.